# Pathogen detection in acellular cerebrospinal fluid: diagnostic insights from a pediatric cohort in Colombia

Jhonny Jesús Patiño Patiño,[1,2] Elber Osorio-Rodríguez,[1,3] Yina Paola García Toscano,[1] Margarita Filott Tamara,[1] Marcio De-Ávila-Arias,[4] Alexander Rodríguez Sanjuán,[5] Jose Luis Villarreal-Camacho,[5] Walter Martínez De la Rosa,[4] Jorge Bilbao Ramírez,[6] Wilfrido Coronell-Rodríguez,[7] Alfonso Bettin Martínez[1,5,8]

**ABSTRACT** The diagnosis of pediatric meningoencephalitis has traditionally relied on the detection of inflammatory markers in cerebrospinal fluid (CSF), particularly pleocytosis, as an indirect indicator of infection. This assumption presumes that microbial invasion of the central nervous system consistently triggers a measurable immune response. In this prospective observational study conducted in Cartagena, Colombia, we enrolled 166 pediatric patients with suspected meningoencephalitis. All patients underwent testing with the FilmArray Meningitis/Encephalitis (FA-M/E) panel for direct pathogen detection in CSF, and results were compared to conventional CSF cytochemical analysis and culture. The FA-M/E panel identified a pathogen in 21 of 166 patients (12.6%), whereas CSF cultures were negative in all cases. Detected pathogens included 16 viruses (76.2%), most frequently Enterovirus and Human Herpesvirus 6 and five bacteria (23.8%), including *Escherichia coli* K1 and *Streptococcus agalactiae*. Remarkably, 100% (21/21) of the FA-M/E-positive samples exhibited complete absence of pleocytosis (0 white blood cells/µL). Additionally, 33.3% (7/21) showed entirely normal CSF biochemistry. Among these cases with a normal CSF profile, the panel identified two clinically significant bacterial infections and five viral infections that would have otherwise gone undetected. These findings demonstrate that traditional CSF parameters are poor predictors of infection in patients with a positive molecular result. Routine reliance on cytochemical analysis alone may therefore delay diagnosis and treatment. Our results support the use of molecular diagnostics as a frontline tool, even in the absence of classical inflammatory CSF markers.

**IMPORTANCE** This study challenges the long-standing paradigm that abnormal cerebrospinal fluid (CSF) findings are required to justify molecular testing in pediatric meningoencephalitis. We show that critical central nervous system (CNS) infections can present with entirely normal CSF profiles, including cases caused by high-risk pathogens such as *E. coli* K1 and *Streptococcus agalactiae*. By highlighting the limitations of pleocytosis and biochemical markers in detecting early CNS infection, our data underscore the importance of incorporating molecular diagnostics into routine clinical evaluation, particularly for high-risk pediatric populations. Failure to do so may result in missed or delayed diagnoses and suboptimal treatment.

**KEYWORDS** meningitis, FilmArray, cerebrospinal fluid, pleocytosis, *E. coli* K1, *S. agalactiae*

Meningoencephalitis (ME), an inflammatory disease of the central nervous system (CNS), carries a heavy burden of mortality and long-term neurological sequelae, particularly in the pediatric population (1) Survivors often face lifelong disabilities, including hearing loss, cognitive impairment, and motor deficits, imposing a devastating

**Peer Reviewer** Julia Piwoz, Montefiore Medical Center, Bronx, New York, USA

Address correspondence to Alfonso Bettin Martínez, abettin@unimetro.edu.co.

The authors declare no conflict of interest.

toll on patients and healthcare systems (2). In Latin America, despite vaccination efforts, bacterial meningitis remains a significant threat, compounded by phenomena such as serotype replacement in *Streptococcus pneumoniae* and rising antimicrobial resistance (3). The cornerstone of ME diagnosis has historically been the analysis of cerebrospinal fluid (CSF), where the presence of inflammatory markers such as pleocytosis, elevated protein, and low glucose is interpreted as a proxy for microbial infection (4). This diagnostic paradigm, however, assumes that microbial invasion of the CNS invariably triggers a robust and immediate inflammatory response. This assumption overlooks the complex molecular interplay between pathogen virulence factors and the host's innate immune response. Conventional diagnostic methods, such as bacterial culture, are not only slow but also suffer from low sensitivity, a problem exacerbated by the common and necessary practice of administering empirical antibiotics prior to lumbar puncture (5).

Previous studies have reported low sensitivity of conventional CSF culture in pediatric CNS infections, often yielding no bacterial growth even in clinically suspected cases (6, 7). The advent of syndromic multiplex PCR panels, such as the FilmArray Meningitis/Encephalitis (FA-M/E) panel, represents a technological leap, enabling the direct and rapid detection of pathogen nucleic acids from a small CSF volume (7). This technology allows us to bypass the reliance on the host's secondary inflammatory response and directly interrogate the presence of the microbe itself. While the clinical utility of these panels is increasingly recognized, their findings also present an opportunity to investigate fundamental questions of molecular pathogenesis *in vivo*.

Recent pediatric studies suggest a clinical mismatch between the presence of a pathogen in the CNS and the classic CSF findings of meningitis (8). This phenomenon, often termed acellular or paucicellular meningitis, may occur when pathogens employ sophisticated molecular strategies to invade the CNS while evading or delaying detection by the host immune system (9). This is particularly relevant in neonates and infants, whose immature immune systems may mount atypical responses (10). If clinical decisions, including the use of advanced molecular diagnostics, are gated by traditional inflammatory markers, a significant number of infections may be missed during a critical therapeutic window.

This study employs the high sensitivity of the FA-M/E panel in a prospective pediatric cohort in Cartagena, Colombia, to examine this clinical mismatch between pathogen presence and inflammatory response in the CSF. Our goal was to determine how this molecular tool performs in clinical practice and whether it can reveal infections that escape traditional diagnostic markers, underscoring its value as a first-line approach in the evaluation of suspected CNS infections.

## RESULTS

### Patient demographics and FilmArray panel yield

A final cohort of 166 pediatric patients with suspected CNS infection was enrolled (Fig. 1). The median age was 1 year (interquartile range [IQR]: 27.3 days–1.64 years), with 51.8% of patients being infants (≤1 year old). The FA-M/E panel identified a pathogen in 21 of 166 samples (12.6%). Consistent with prior reports, all CSF cultures in our cohort were negative, while the FilmArray M/E panel identified pathogens in 21 cases. In contrast, standard bacterial and fungal CSF cultures were negative in 100% of the 166 samples, including all 21 cases that were positive by the FA-M/E panel. Of the 21 positive detections, 16 (76.1%) were viral and 5 (23.8%) were bacterial. No fungal pathogens were detected. The most frequently identified pathogens were Enterovirus and Human Herpesvirus 6 (HHV-6). Two cases of co-detection were observed: *Streptococcus agalactiae* with HHV-6 and *Escherichia coli* K1 with Enterovirus. The distribution of detected microorganisms is detailed in Fig. 2.

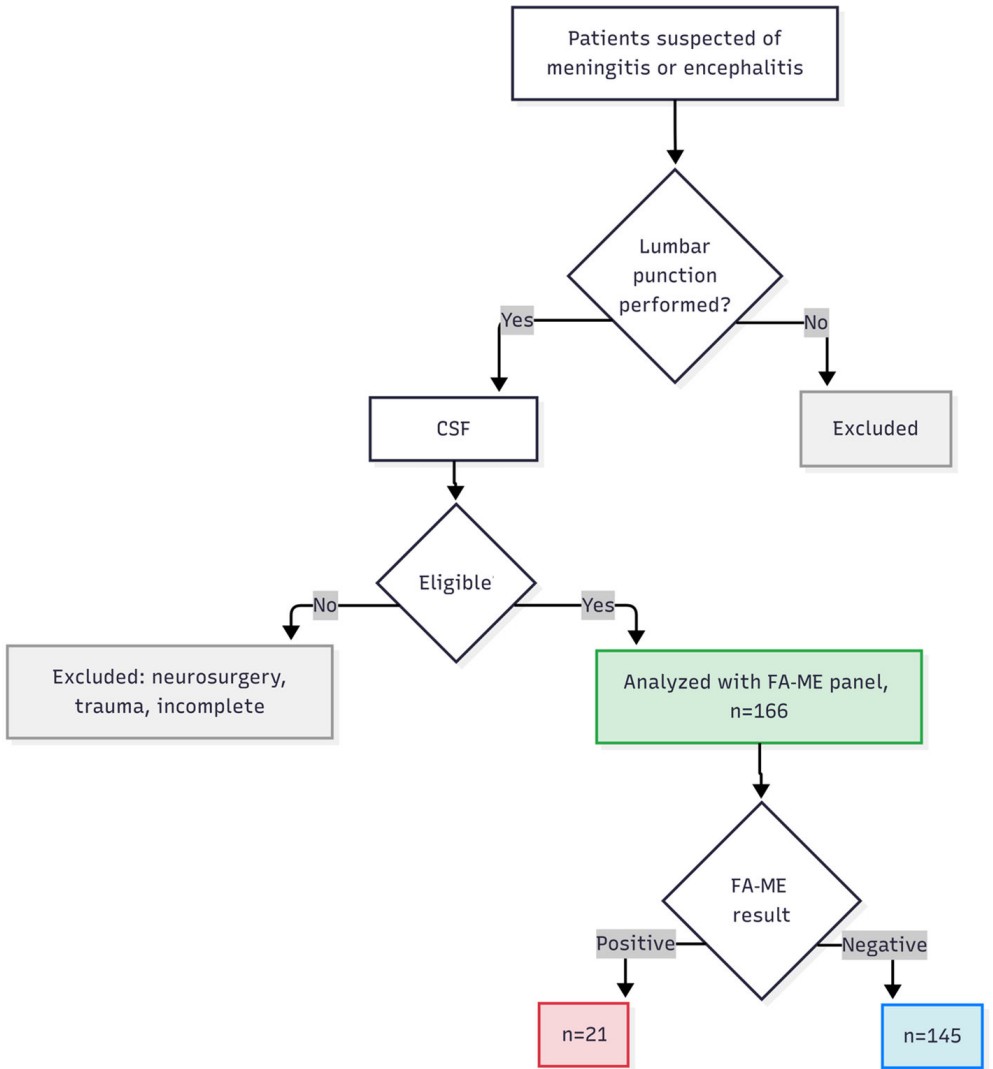

**FIG 1** Flowchart of patient inclusion and classification. Patients with a clinical suspicion of meningitis or encephalitis were initially screened. After lumbar puncture, cerebrospinal fluid (CSF) samples were evaluated for eligibility. Exclusion criteria included recent neurosurgery, traumatic brain injury, repeated samples, or incomplete clinical data. A total of 166 patients were included and underwent analysis using the FilmArray Meningitis/Encephalitis (FA-ME) panel. Based on the FA-ME results, patients were categorized as positive ($n = 21$) or negative ($n = 145$).

## Clinical outcomes

While baseline clinical and demographic characteristics were similar between the FA-M/E-positive and FA-M/E-negative groups (Table 1), patients with a confirmed pathogen had significantly longer hospital stays (median 6 vs 3 days, $P = 0.006$) and ICU stays (median 13 vs 7 days, $P = 0.029$). This suggests that the panel was effective at identifying patients with a more severe underlying disease requiring more intensive care (Fig. 3). While baseline clinical and demographic characteristics were similar between the FA-M/E-positive and FA-M/E-negative groups (Table 1), patients with a confirmed pathogen had significantly longer hospital stays (median 6 vs 3 days, $P = 0.006$) and ICU stays (median 13 vs 7 days, $P = 0.029$). This suggests that the panel was effective at identifying patients with a more severe underlying disease requiring more intensive care (Fig. 3). However, given the small sample size ($n = 21$), these outcome analyses should be considered exploratory.

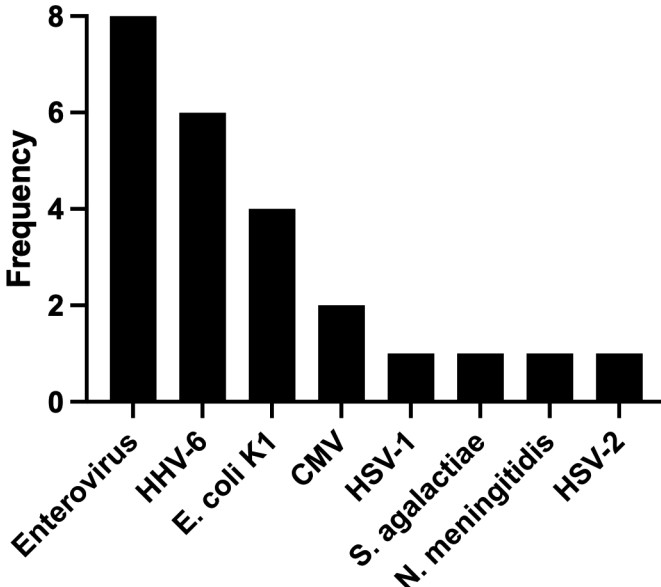

**FIG 2** Etiological distribution of pathogens detected by the FilmArray M/E panel. The bar chart illustrates the frequency of each unique pathogen identified across the 21 positive patient samples. Enterovirus and Human Herpesvirus 6 (HHV-6) were the most frequently detected agents. The total number of pathogens identified ($n = 24$) is greater than the number of positive patients ($n = 21$) due to three cases of co-detection (*Streptococcus agalactiae* with HHV-6, *Escherichia coli* K1 with Enterovirus, and Enterovirus with HHV-6).

## Normal CSF profiles are confirmed by FA-M/E with CNS infections

The central finding of this study showed a clinical mismatch between the molecular confirmation of a CNS infection and the presence of traditional inflammatory markers in the CSF.

### *Absence of pleocytosis*

None of the 21 patients (0%) with a positive FA-M/E result exhibited pleocytosis (detectable white blood cells) in their CSF. Similarly, none had an abnormal (turbid) CSF appearance or elevated opening pressure (Table 2).

### *Normal CSF biochemistry*

A significant proportion of positive cases with FA-M/E also had normal CSF biochemistry. Seven of the 21 positive patients (33.3%) presented with both normal glucose and normal protein levels. Within this subgroup with a normal CSF profile, the FA-M/E panel identified two critical bacterial infections (*E. coli* K1 and *S. agalactiae*) and five viral infections that would have otherwise been missed (Table 2). To further characterize these acellular profiles, we next evaluated individual and combined cytochemical parameters.

## CSF cytochemical findings associated with FilmArray M/E results

Evaluation of individual CSF cytochemical parameters (Table 2) revealed that none of the 21 FilmArray‐positive patients exhibited abnormal appearance, elevated opening pressure, or pleocytosis (0/21 [0%]); these proportions did not differ significantly from FilmArray‐negative cases ($P > 0.05$ for all). Regarding glucose levels, 120/166 (72.3%) of all patients had normal CSF glucose, and 28/166 (16.9%) had hypoglycorrhachia. Among FilmArray‐positive samples with normal glucose, 3/21 (14.3%) were bacterial, 10/21 (47.6%) viral, and 1/21 (4.8%) mixed bacterial–viral detections; in those with hypoglycorrhachia, 1/21 (4.8%) were bacterial; 4/21 (19.0%) were viral; and 1/21 (4.8%) were mixed.

**TABLE 1** Baseline demographic, clinical, laboratory, and outcome characteristics of the cohort by FilmArray M/E panel result[f,g]

| Variables | Total (n = 166) | FilmArray positive (n = 21) | FilmArray negative (n = 145) | P value |
|---|---|---|---|---|
| Sociodemographics, n (%) | | | | |
| Male | 94 (56.6) | 10 (47.6) | 84 (57.9) | 0.373[a] |
| Female | 72 (43.4) | 11 (52.4) | 61 (42.1) | |
| ≤1 year | 86 (51.8) | 12 (57.1) | 74 (51.0) | 0.601[a] |
| >1 year | 80 (48.2) | 9 (42.9) | 71 (49.0) | |
| Signs and symptoms, n (%) | | | | |
| Fever | 147 (88.6) | 20 (95.2) | 127 (87.6) | 0.472[a] |
| Tachycardia | 95 (57.2) | 8 (38.1) | 87 (60.0) | 0.069[a] |
| Tachypnea | 25 (15.1) | 5 (23.8) | 20 (13.8) | 0.322[a] |
| Neurological alterations[d] | 42 (25.3) | 6 (28.6) | 36 (24.8) | 0.712[a] |
| Gastrointestinal symptoms | 23 (13.9) | 3 (14.3) | 20 (13.8) | 1.000[b] |
| Laboratory at admission, median (IQR) | | | | |
| Leukocytes (×10³/mm³) | 10,400 (7,750–13,600) | 9,000 (6,200–11,400) | 10,600 (7,800–14,000) | 0.089[c] |
| Neutrophils (%) | 54.4 (37–64) | 55.3 (37.0–62.8) | 54.0 (37–64) | 0.951[c] |
| Lymphocytes (%) | 36.7 (27.1–48.8) | 34.5 (29.8–45.6) | 37.2 (27–49) | 0.541[c] |
| Platelets (no./mm³) | 352,000 (258,750–449,750) | 355,000 (277,000–458,000) | 351,000 (250,000–446,000) | 0.820[c] |
| Complicated, n (%) | | | | |
| Cardiovascular | 9 (5.4) | 0 (0) | 9 (6.2) | 0.605[b] |
| Respiratory | 26 (15.7) | 4 (19.0) | 22 (15.2) | 0.747[b] |
| Renal | 3 (1.8) | 0 (0) | 3 (2.1) | 1.000[b] |
| Neurological | 4 (2.4) | 1 (4.8) | 3 (2.1) | 0.421[b] |
| Length of stay, days (IQR) | | | | |
| Hospitalization | 4 (2.0–7.5) | 6 (4–12) | 3 (2–7) | 0.006[c] |
| ICU | 7 (4–12) | 13 (10.3–18.0)[e] | 7 (3–11) | 0.029[c] |
| Clinical outcomes, n (%) | | | | |
| Neurological sequelae | 18 (10.8) | 2 (9.5) | 16 (11.0) | 1.000[b] |
| Death | 4 (2.4) | 0 (0) | 4 (2.8) | 1.000[b] |

[a]Fisher's exact test.
[b]For categorical variables and the Mann–Whitney U test.
[c]For continuous variables, a P value of <0.05 was considered statistically significant.
[d]Indicates convulsion or Glasgow Coma Scale score of ≤13.
[e]Indicates hydrocephalus.
[f]FA-M/E, FilmArray Meningitis/Encephalitis; GCS, Glasgow Coma Scale; ICU, intensive care unit; IQR, interquartile range; WBC, white blood cells.
[g]Data are presented as n (%) for categorical variables and median (IQR) for continuous variables. Group comparisons between FilmArray-positive and FilmArray-negative patients were conducted using the chi-square test, Fisher's exact test for categorical variables, and the Mann–Whitney U test for continuous variables.

For protein, 82/166 (49.4%) of all patients had normal CSF protein despite 2/21 (9.5%) bacterial and 6/21 (28.6%) viral detections among FilmArray positives. Although 62/166 (37.3%) displayed hyperproteinorrhachia, this did not differ significantly by FilmArray status (11/21 [52.4%] positive vs 51/145 [35.2%] negative, P = 0.128).

Combined parameter analysis (Table 2) showed that normal glucose and normal protein was the most frequent phenotype (67/166, 40.4%), with no significant difference between FilmArray‐positive and FilmArray‐negative groups (7/21 [33.3%] vs 60/145 [41.4%], P = 0.482). Among FilmArray positives with this combination, 2/21 (9.5%) were bacterial and 5/21 (23.8%) were viral. Hyperproteinorrhachia with normal glucose occurred in 37/166 (22.3%) of all patients, of which 5/21 (23.8%) were viral and 1/21 (4.8%) were bacterial in the positive cohort. Finally, hypoglycorrhachia combined with hyperproteinorrhachia was observed in 20/166 (12.0%) overall, but only 3/21 (14.3%) of positives were viral; 1/21 (4.8%) were bacterial; and 1/21 (4.8%) were mixed. This combination also showed no significant difference by FilmArray status (5/21 [23.8%] vs 15/145 [10.3%], P = 0.141).

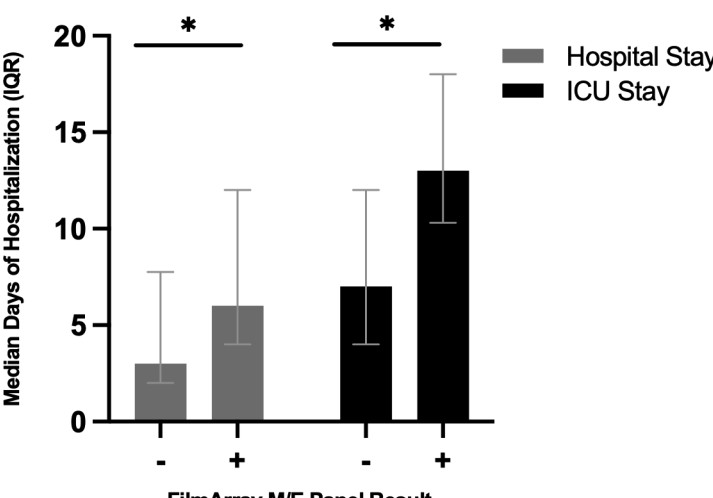

## Impact of FilmArray Result on Length of Stay

**FIG 3** Hospital and ICU length of stay stratified by FilmArray M/E Result. Boxplots depict the median and interquartile range of total hospital stay (left) and intensive care unit (ICU) stay (right) in days for patients with a positive FilmArray M/E panel ($n = 21$) vs those with a negative result ($n = 145$). Whiskers extend to the most extreme data points not considered outliers. Statistical significance between groups was assessed using the Mann–Whitney $U$ test; $P < 0.05$ (*) are indicated above each comparison.

## Diagnostic performance of CSF parameters

The overall diagnostic performance of standard CSF cytochemical parameters for predicting a positive FilmArray M/E result was uniformly poor (Table 2). No single marker or combination demonstrated sufficient sensitivity or positive predictive value (PPV)

**TABLE 2** Individual and combined cerebrospinal fluid cytochemical findings by FilmArray M/E panel result[c,d]

| Variables | Total ($n = 166$) | FilmArray positive | | | FilmArray negative ($n = 145$) | P value |
|---|---|---|---|---|---|---|
| | | Bacteria ($n = 4$) | Virus ($n = 15$) | Bacteria–virus ($n = 2$) | | |
| Individual biomarkers, n (%) | | | | | | |
| Appearance (abnormal) | 11 (6.6) | 0 | 0 | 0 | 11 (7.6) | 0.362[a] |
| Elevated opening pressure | 15 (9.0) | 0 | 0 | 0 | 15 (10.3) | 0.221[a] |
| Leukocytes | 7 (4.2) | 0 | 0 | 0 | 7 (4.8) | 0.597[a] |
| Normal CSF glucose | 120 (72.3) | 3 (14.3) | 10 (47.6) | 1 (4.8) | 106 (73.1) | 0.538[b] |
| Hypoglycorrhachia | 28 (16.9) | 1 (4.8) | 4 (19.0) | 1 (4.8) | 22 (15.2) | 0.130[a] |
| Normal CSF protein | 82 (49.4) | 2 (9.5) | 6 (28.6) | 0 (0) | 74 (51.0) | 0.268[b] |
| Hyperproteinorrhachia | 62 (37.3) | 1 (4.8) | 8 (38.1) | 2 (9.5) | 51 (35.2) | 0.128[b] |
| Hypoproteinorrhachia | 22 (13.3) | 1 (4.8) | 1 (4.8) | 0 (0) | 20 (13.8) | 0.590[b] |
| Combined biomarkers, n (%) | | | | | | |
| Normal glucose + normal protein | 67 (40.4) | 2 (9.5) | 5 (23.8) | 0 (0) | 60 (41.4) | 0.482[b] |
| Hypoglycorrhachia + normal protein | 8 (4.8) | 0 (0) | 1 (4.8) | 0 (0) | 7 (4.8) | 1.000[a] |
| Hyperproteinorrhachia + normal glucose | 37 (22.3) | 0 (0) | 5 (23.8) | 1 (4.8) | 31 (21.4) | 0.574[a] |
| Normal glucose + hypoproteinorrhachia | 16 (9.6) | 1 (4.8) | 0 (0) | 0 (0) | 15 (10.3) | 0.696[a] |
| Hypoglycorrhachia + hypoproteinorrhachia | 8 (4.8) | 0 (0) | 1 (4.8) | 0 (0) | 7 (4.8) | 1.000[a] |
| Hypoglycorrhachia + hyperproteinorrhachia | 20 (12.0) | 1 (4.8) | 3 (14.3) | 1 (4.8) | 15 (10.3) | 0.141[a] |

[a]Chi-square test.
[bc]Fisher's exact test.Data are presented as $n$ (%) for each CSF parameter. Comparisons between FilmArray‐positive subgroups (bacterial, viral, and bacterial–viral) and FilmArray‐negative patients were performed using the chi‐square test or Fisher's exact test; a $P$ value of <0.05 was considered statistically significant.
[d]CSF, cerebrospinal fluid; FA-M/E, FilmArray Meningitis/Encephalitis.

**TABLE 3** Diagnostic performance of individual and combined CSF cytochemical findings for the FA-M/E panel[a,b]

| Variables | Sensitivity | | Specificity | | Positive predictive value | | Negative predictive value | |
|---|---|---|---|---|---|---|---|---|
| | (TP/TP + FN) | % | (TN/FP + TN) | % | (TP/TP + FP) | % | (TN/TN + FN) | % |
| Individual biomarkers | | | | | | | | |
| Appearance (abnormal) | 0/21 | 0 | 134/145 | 92.4 | 0/11 | 0 | 134/155 | 86.5 |
| Elevated opening pressure | 0/21 | 0 | 130/145 | 89.7 | 0/15 | 0 | 130/151 | 86.1 |
| Leukocytes | 0/21 | 0 | 138/145 | 95.2 | 0/7 | 0 | 138/159 | 86.8 |
| Normal CSF glucose | 14/21 | 66.7 | 39/145 | 26.9 | 14/120 | 11.7 | 39/46 | 84.8 |
| Hypoglycorrhachia | 6/21 | 28.6 | 123/145 | 84.8 | 6/28 | 21.4 | 123/138 | 89.1 |
| Normal CSF protein | 8/21 | 38.1 | 71/145 | 49 | 8/82 | 9.8 | 71/84 | 84.5 |
| Hyperproteinorrhachia | 11/21 | 52.4 | 94/145 | 64.8 | 11/62 | 17.7 | 94/104 | 90.4 |
| Hypoproteinorrhachia | 2/21 | 9.5 | 125/145 | 86.2 | 2/22 | 9.1 | 125/144 | 86.8 |
| Combined biomarkers | | | | | | | | |
| Normal glucose + normal protein | 7/21 | 33.3 | 85/145 | 58.6 | 7/67 | 10.4 | 85/99 | 85.9 |
| Hypoglycorrhachia + normal protein | 1/21 | 4.8 | 138/145 | 95.2 | 1/8 | 12.5 | 138/158 | 87.3 |
| Hyperproteinorrhachia + normal glucose | 6/21 | 28.6 | 114/145 | 78.6 | 6/37 | 16.2 | 114/129 | 88.4 |
| Normal glucose + hypoproteinorrhachia | 1/21 | 4.8 | 130/145 | 89.7 | 1/16 | 6.3 | 130/150 | 86.7 |
| Hypoglycorrhachia + hypoproteinorrhachia | 1/21 | 4.8 | 138/145 | 95.2 | 1/8 | 12.5 | 138/158 | 87.3 |
| Hypoglycorrhachia + hyperproteinorrhachia | 5/21 | 23.8 | 130/145 | 89.7 | 5/20 | 25 | 130/146 | 89 |

[a]Sensitivity (TP/[TP + FN]), specificity (TN/[FP + TN]), positive predictive value (PPV = TP/[TP + FP]) and negative predictive value (NPV = TN/[TN + FN]) are shown as both fraction and percentage. A $P$ value of <0.05 was considered statistically significant.
[b]CSF, cerebrospinal fluid; FA-M/E, FilmArray Meningitis/Encephalitis; FP, false positive; FN, false negative; TP, true positive; TN, true negative.

to reliably guide molecular testing (Table 3). Evaluation of individual markers (Table 3) showed that normal CSF glucose offered the highest sensitivity at 66.7%, followed by elevated protein at 52.4%. However, their specificity was modest. The most specific findings included leukocytosis (95.2%), abnormal appearance (92.4%), elevated opening pressure (89.7%), and low protein (86.2%), all of which suffered from near-zero sensitivity. All individual PPVs remained below 22%, despite negative predictive values (NPVs) exceeding 84%.

When examining combined parameters (Table 3), the pairing of normal glucose with normal protein achieved the greatest sensitivity (33.3%), followed by elevated protein with normal glucose (28.6%). Combinations involving hypoglycorrhachia, whether paired with normal or low protein, yielded a sensitivity of just 4.8% while maintaining high specificity (>95%). Yet their PPVs did not exceed 17%, and NPVs ranged only between 85.9% and 89.0%. These data confirm that reliance on routine CSF cytochemistry to triage patients for multiplex PCR is both insensitive and unspecific, risking missed diagnoses of CNS infection.

## DISCUSSION

This study reveals a critical unexpected mismatch in pediatric meningoencephalitis: the presence of a pathogen in the CNS, confirmed by sensitive molecular methods, frequently occurs in the absence of the canonical host inflammatory response in the CSF. Our finding that 100% of molecularly confirmed infections were acellular (lacking pleocytosis) challenges the widely accepted clinical dogma that a normal CSF analysis can reliably exclude significant CNS infection. To confirm the clinical relevance of these acellular detections, we performed a retrospective clinical adjudication (RCA) on all 21 positive cases (Table S1). The RCA classified most of these detections as clinically significant definite or probable infections, supporting the conclusion that traditional CSF cytochemical parameters are poor predictors of molecularly confirmed CNS infection. We propose that this discrepancy is not due to diagnostic error but rather highlights the ability of certain pathogens to cause CNS infection without triggering an early inflammatory response, emphasizing the clinical value of molecular diagnostics like the

FilmArray panel in guiding both diagnosis and timely therapeutic decisions, even when conventional CSF findings appear normal.

Our finding that 100% of molecularly confirmed infections were acellular differs substantially from the landmark multicenter evaluation of the FA-M/E panel by Leber et al. (7), as well as other large pediatric cohorts, where pleocytosis is the expected finding (11, 12). For instance, in a large pediatric study utilizing the same panel, pleocytosis was observed in most bacterial cases (87%) and in a significant portion of viral cases (54.4%) (11). Similarly, another study reported the absence of pleocytosis in only 11% of their positive cases (6), a figure that stands in sharp opposition to the 100% observed in our cohort. A national study from the Colombian Pediatric Encephalitis Network (13) also reported low CSF leukocyte counts as a risk factor for pediatric intensive care unit admission, suggesting that even in the absence of CSF abnormalities, some infections may still require intensive management. Together, these findings highlight the unusual nature of our cohort and suggest that the absence of CSF inflammation in the presence of infection may be more common and more clinically relevant than previously recognized, particularly in specific epidemiological or age-defined contexts. This discrepancy between our findings and the established literature is precisely what underscores the unique nature of our results and compels a deeper investigation into the underlying pathogenic mechanisms, rather than dismissing it as a mere diagnostic anomaly (6, 7, 11).

The detection of bacteria like *E. coli* K1 and *Streptococcus agalactiae* in patients with normal CSF parameters is particularly alarming but explainable. In our cohort, FilmArray results were available within one to two hours, whereas conventional CSF cultures remained negative even after seventy-two hours of incubation. This rapid turnaround allowed clinicians to de-escalate from broad‑spectrum empiric antibiotics such as meropenem and vancomycin to targeted agents. Among the five bacterial infections detected, four involved *Escherichia coli* K1 and one involved *Streptococcus agalactiae*. Patients with *E. coli* K1 had a median hospital stay of 12.5 days (range 7–43), while the single *S. agalactiae* case had a 7‑day stay. Multiplex PCR confirmation prompted a switch to ceftriaxone within a median of 24 hours of admission. Early adjustment of therapy reduced unnecessary antimicrobial exposure and enabled more precise dosing and duration. These practice changes may decrease the risk of drug-related toxicity, limit the emergence of resistance, and shorten overall hospital stays. Our findings support adopting the FA-M/E panel as a first-line diagnostic tool to guide timely, pathogen-directed therapy in pediatric CNS infections.

Mechanistically, these clinical explanations reflect how these pathogens have evasion strategies to breach the blood-brain barrier (BBB) (9) *Escherichia coli* K1, a leading cause of neonatal meningitis, utilizes its K1 polysialic acid capsule as a molecular shield. This capsule mimics host structures, such as the neural cell adhesion molecule, effectively camouflaging the bacterium from the host's complement system and phagocytes (14). *E. coli* K1 employs a host cell-mediated transport-like entry at the cellular level, which explains how the bacterium can establish a foothold in the CNS before a significant number of neutrophils are recruited into the CSF, thus accounting for the acellular CSF observed in our patients (15).

Similarly, *Streptococcus agalactiae* (Group B Streptococcus [GBS]) possesses a complex molecular toolkit for CNS invasion. Its pathogenesis relies on a multistep process involving adherence, invasion, and evasion of host defenses (16, 17). GBS expresses an array of surface proteins, such as the serine-rich repeat proteins and pili, which mediate high-affinity binding to host extracellular matrix components like fibrinogen and laminin, anchoring the bacterium to the endothelial cell surface (17). Following adhesion, GBS can trigger its own uptake into host cells or secrete toxins like the β-hemolysin/cytolysin, a pore-forming toxin that facilitates tissue damage and penetration of host cell barriers, including the BBB (16). This molecularly driven invasion allows GBS to cross the BBB directly, again preceding the development of a massive inflammatory infiltrate that would be reflected as pleocytosis in the CSF (18). Previous studies have

reported occasional false-positive FilmArray results for GBS, including one of the reports cited in our study. Therefore, GBS detections should be interpreted cautiously and, when feasible, confirmed by culture or an independent molecular method to ensure diagnostic accuracy (6, 7).

The viral pathogens detected, particularly Enterovirus, also utilize neuroinvasion strategies that can explain the lack of initial CSF inflammation (19). Enteroviruses are well-documented neurotropic viruses that can invade the CNS via retrograde axonal transport from peripheral sites, such as the gut or muscle tissue (20). By traveling within nerve axons, the virus can effectively bypass the bloodstream and the meningeal immune surveillance, reaching the CNS without triggering an early, widespread meningeal inflammation (21, 22). The detection of enteroviral RNA in the CSF is therefore a direct indicator of CNS involvement (23–25). This supports the use of molecular testing for early diagnosis, even in patients with normal CSF profiles (26).

The case of HHV-6 is particularly complex and inherently molecular. HHV-6 can exist in a latent state, reactivate, or even be chromosomally integrated (ciHHV-6) (27, 28). A positive PCR result for HHV-6 in the CSF therefore requires careful clinical correlation, as it may represent true encephalitis, clinically irrelevant reactivation, or the presence of ciHHV-6 in host cells shed into the CSF (29). Conventional CSF parameters are not informative in differentiating these molecular states, making viral load quantification essential to clarify the pathogen's role in a given patient (30). The absence of quantitative viral load or serologic confirmation is a limitation of our study, and detection of HHV-6 DNA in CSF should be interpreted with caution, as it may reflect latent infection or reactivation rather than acute neuroinfection.

## Implications for diagnostic paradigms and future research

Our findings have implications for both clinical practice and future research. Clinically, they provide strong evidence that diagnostic algorithms for ME should not be gated by CSF parameters. Restricting the use of powerful molecular tools like the FA-M/E panel to patients with pleocytosis is an unsafe practice that will lead to missed or delayed diagnoses. Diagnostic stewardship should instead be guided by clinical risk factors (e.g., age <3 months, immunocompromised status, and signs of encephalitis) that reflect a higher pre-test probability of infection, irrespective of initial CSF findings (31, 11).

Our data also challenge current practices that limit the use of multiplex PCR testing to patients with abnormal CSF profiles. We showed that restricting the FA-M/E panel based on conventional cytochemical parameters would have delayed the diagnosis of CNS infection in all 21 positive cases in our cohort. Supporting this concern, a previous study found that patients with confirmed CNS infection but normal CSF parameters had worse outcomes, likely due to delays in antimicrobial treatment compared to those with abnormal CSF findings (11). These results reinforce the need to decouple molecular testing decisions from CSF biochemistry alone, particularly in high-risk populations.

From a research perspective, this study demonstrates the power of applying sensitive molecular diagnostics in a clinical setting to uncover fundamental insights into microbial pathogenesis (32). The normal CSF in these infected patients is not a sign of no infection, but rather a sign of a pathogen that has successfully employed its molecular virulence machinery to enter the CNS undetected by the initial wave of immune response (32). Future studies should rely on these findings by coupling syndromic testing with quantitative PCR, transcriptomics, and proteomics on CSF samples to further dissect the specific host–pathogen interactions and gene expression profiles that define these early, acellular stages of CNS infection (33).

## Limitations

This study has several limitations. Its single-center, observational design may introduce selection bias and limit applicability across different clinical settings. However, the central finding regarding the complete mismatch between molecular detection and CSF inflammation represents a pronounced and biologically plausible signal that warrants

validation in larger, multicenter cohorts. Additionally, we did not use an independent confirmatory method to verify the FilmArray results which could help differentiate between low-level contamination, latency (especially for HHV-6), and active, high-level replication.

In our study, the limited sample size may restrict the generalizability of the findings. Moreover, the relatively small number of FilmArray-positive cases ($n = 21$) limited the statistical power to assess clinical outcomes such as hospital or ICU length of stay. These analyses should therefore be interpreted as exploratory. In addition, no further confirmatory tests were performed to validate molecular results, and therefore, clinical adjudication of infectious etiology was not always possible.

The normal CSF in these infected patients is not a sign of no infection, but rather a sign of a pathogen that has successfully employed its molecular virulence machinery to enter the CNS undetected. This creates opportunities for new avenues for research. Future studies should couple syndromic testing with quantitative molecular methods and next-generation technologies like CSF proteomics and metabolomics. These approaches could identify novel, non-cellular biomarkers (e.g., specific host proteins or microbial metabolites) that signal these early, acellular stages of CNS infection, leading to even earlier and more precise diagnostics (32, 33).

## Conclusion

In this prospective pediatric cohort, we found a complete disconnect between molecular pathogen detection and the presence of CSF pleocytosis, which is traditionally considered the hallmark of meningitis. Rather than indicating a diagnostic failure, this pattern highlights the clinical relevance of early immune evasion strategies used by pathogens such as *Escherichia coli* K1 and *Streptococcus agalactiae*, which may cross the blood-brain barrier without triggering an immediate inflammatory response. These findings underscore the limitations of relying solely on CSF cytochemical markers to guide diagnostic decisions. Our results support a shift in clinical strategy: molecular tools such as the FilmArray M/E panel should be considered early in the diagnostic process based on clinical suspicion, even when CSF parameters appear normal. This approach improves diagnostic accuracy, supports more targeted antimicrobial decisions, and may lead to better outcomes in pediatric CNS infections. Beyond its clinical utility, direct pathogen detection also offers a valuable opportunity to understand the early, acellular phases of CNS infection and should be further explored through advanced molecular and proteomic techniques.

## MATERIALS AND METHODS

### Study design and setting

We conducted a prospective, observational, analytical study at the Hospital Infantil Napoleón Franco Pareja, a tertiary pediatric referral center in Cartagena, Colombia, over the course of 1 year. Our primary objective was to compare the clinical utility and diagnostic performance of the FA-M/E panel with conventional CSF analyses in neonatal and pediatric patients presenting with suspected CNS infection.

### Patient selection

All neonates and pediatric patients (0–18 years) who underwent lumbar puncture (LP) for suspicion of meningitis or encephalitis were included. The procedure was performed in patients presenting with fever plus one or more of the following: seizures, altered level of consciousness (Glasgow Coma Scale score ≤13), new focal neurological deficit, or clinical meningeal signs (Table S1). Exclusion criteria were prior neurosurgical intervention (e.g., recent head trauma or ventricular shunt placement), incomplete clinical records, and repeat CSF sampling from the same episode. After applying these criteria, 166 unique patients were included in the final analysis.

## Sample collection and laboratory testing

CSF was collected via standard aseptic LP. A minimum of 500 µL per sample was divided as follows:

For the FA-M/E panel, 200 µL processed on the fully automated FilmArray system (BioFire Diagnostics, Salt Lake City, UT, USA), which multiplexes PCR to detect 14 common pathogens (six bacteria, seven viruses, and one fungus). Testing adhered to the manufacturer's biosafety and operational guidelines and was available 24/7. For the conventional analyses, the remaining volume was used for cytochemical studies (cell count, glucose, protein, and opening pressure) and bacterial/fungal cultures in the institutional microbiology laboratory.

## Data collection and definitions

We extracted demographic data (age and sex), clinical features (fever, tachycardia, tachypnea, seizures, altered consciousness, and gastrointestinal symptoms), complications (cardiovascular, respiratory, renal, and neurological), and outcomes (neuroimaging abnormalities, ICU admission, length of stay, and mortality) from electronic medical records. Hematologic parameters included absolute leukocyte and platelet counts and differential percentages of neutrophils and lymphocytes.

CSF parameters were categorized according to institutional laboratory thresholds:

- Appearance: clear vs turbid.
- Opening pressure: elevated vs normal (age adjusted).
- Leukocytes: present vs absent.
- Glucose: normal (37–75 mg/dL), hypoglycorrhachia (≤37 mg/dL), and hyperglycorrhachia (≥75 mg/dL).
- Protein: normal (20–80 mg/dL), hypoproteinorrhachia (≤20 mg/dL), and hyperproteinorrhachia (≥80 mg/dL).

Patients were stratified by FA-M/E result (positive vs negative). Positive cases were further subclassified by pathogen type (bacterial, viral, or mixed). We evaluated individual CSF markers and selected combinations for their association with, and predictive value for, a positive FA-M/E result, as well as their impact on clinical decision-making.

## Statistical analysis

Continuous variables were assessed for normality via the Shapiro–Wilk test; non-parametric data are presented as medians and IQRs (25th–75th percentile) and compared using the Mann–Whitney $U$ test. Categorical variables are expressed as counts and percentages, with group comparisons conducted using the chi-square or Fisher's exact test, as appropriate. Diagnostic performance metrics sensitivity, specificity, PPV, and NPV were calculated for each CSF marker and combination, using the FA-M/E panel result as the reference standard. A two-tailed $P < 0.05$ was considered statistically significant. All analyses were performed in STATA v.17.0 (StataCorp, College Station, TX, USA). Reporting adheres to the Strengthening the Reporting of Observational Studies in Epidemiology guidelines.

## ACKNOWLEDGMENTS

The authors thank the dedicated professionals at the study site, without whom this work would not have been possible. Special thanks go to Ines Martínez, Herando Pinzon, Keterine Palacio from Hospital Infantil Franco Napoleon Pareja, Casa del Niño.

This study was designed by Universidad Metropolitana, Colombia, and was funded by BioFire Diagnostics (RCA# BFD-ME-18-004).

## AUTHOR AFFILIATIONS

[1]Grupo Caribe de Investigación en Enfermedades de Tipo Infeccioso y Resistencia Microbiana, Universidad Metropolitana, Barranquilla, Colombia

[2]Departamento de Medicina Interna, Universidad de Cartagena, Cartagena, Colombia

[3]Department of intensive Medicine, Group Care Medicine, Clínica Iberoamérica., Barranquilla, Colombia

[4]Grupo de investigación en Biomedicina, Facultad de Ciencias básicas biomédicas, Universidad Metropolitana, Barranquilla, Colombia

[5]Grupo de Investigación en Medicina Traslacional, Departamento de Medicina, Universidad Metropolitana, Barranquilla, Colombia

[6]Grupo SANUS VIVENTIUM, Departamento de Investigaciones, Universidad Metropolitana, Barranquilla, Colombia

[7]Facultad de Medicina, Departamento de Pediatría e Infectología, Universidad de Cartagena, Cartagena, Colombia

[8]Grupo de Investigación de Odontología Universidad Metropolitana (GIOUMEB), Barranquilla, Colombia

## AUTHOR ORCIDs

Jhonny Jesús Patiño Patiño http://orcid.org/0000-0002-5764-6326
Elber Osorio-Rodríguez http://orcid.org/0000-0001-9404-5370
Yina Paola García Toscano http://orcid.org/0009-0008-7739-3451
Marcio De-Ávila-Arias http://orcid.org/0000-0001-7637-5106
Alexander Rodríguez Sanjuán http://orcid.org/0000-0001-6424-7254
Walter Martínez De la Rosa http://orcid.org/0000-0003-4106-9357
Alfonso Bettin Martínez http://orcid.org/0000-0002-8335-9929

## AUTHOR CONTRIBUTIONS

Jhonny Jesús Patiño Patiño, Conceptualization, Data curation, Formal analysis, Funding acquisition, Investigation, Methodology, Project administration, Resources, Supervision, Validation, Visualization, Writing – original draft, Writing – review and editing | Elber Osorio-Rodríguez, Conceptualization, Data curation, Investigation, Methodology, Writing – original draft, Writing – review and editing | Yina Paola García Toscano, Conceptualization, Data curation, Formal analysis, Investigation, Methodology, Writing – original draft, Writing – review and editing | Margarita Filott Tamara, Conceptualization, Data curation, Formal analysis, Investigation, Writing – original draft, Writing – review and editing | Marcio De-Ávila-Arias, Conceptualization, Data curation, Formal analysis, Investigation, Visualization, Writing – original draft, Writing – review and editing | Alexander Rodríguez Sanjuán, Conceptualization, Data curation, Formal analysis, Investigation, Visualization, Writing – original draft, Writing – review and editing | Jose Luis Villarreal-Camacho, Conceptualization, Data curation, Writing – original draft, Writing – review and editing | Walter Martínez De la Rosa, Conceptualization, Data curation, Writing – original draft, Writing – review and editing | Jorge Bilbao Ramírez, Conceptualization, Data curation, Writing – original draft, Writing – review and editing | Wilfrido Coronell-Rodríguez, Conceptualization, Data curation, Formal analysis, Visualization, Writing – original draft, Writing – review and editing | Alfonso Bettin Martínez, Conceptualization, Data curation, Formal analysis, Funding acquisition, Investigation, Methodology, Project administration, Supervision, Validation, Visualization, Writing – original draft, Writing – review and editing

## ETHICS APPROVAL

The study protocol was granted ethical approval by the Institutional Research Ethics Committee, and written informed consent was obtained from all legal guardians.

## ADDITIONAL FILES

The following material is available online.

### Supplemental Material

**Table S1 (Spectrum02122-25-s0001.docx).** Clinical profile, detected pathogens, and retrospective adjudication of 21 pediatric patients with positive FA-M/E results.

### Open Peer Review

**PEER REVIEW HISTORY (review-history.pdf).** An accounting of the reviewer comments and feedback.

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
