## [Reviewer comments · Microbiology Spectrum]

Microbiology Spectrum

Pathogen Detection in Acellular Cerebrospinal Fluid: Diagnostic Insights from a Pediatric Cohort in Colombia

Jhonny Patiño Patiño, Elber Osorio-Rodríguez, Yina García Toscano, Margarita Filott Tamara, Marcio De-Ávila-Arias, Alexander Rodríguez Sanjuán, Jose Villarreal-Camacho, Walter Martínez De la Rosa, Jorge Bilbao Ramírez, Wilfrido Coronell-Rodríguez, and Alfonso Bettin Martínez

Corresponding Author(s): Alfonso Bettin Martínez, Universidad Metropolitana de Barranquilla

Review Timeline:

Submission Date:	July 14, 2025
Editorial Decision:	September 8, 2025
Revision Received:	October 27, 2025
Accepted:	October 29, 2025

Editor: Eleanor Powell

Reviewer(s): Disclosure of reviewer identity is with reference to reviewer comments included in decision letter(s). The following individuals involved in review of your submission have agreed to reveal their identity: Julia Piwoz (Reviewer #1)

Transaction Report:

DOI: <https://doi.org/10.1128/spectrum.02122-25>

Re: Spectrum02122-25 (Pathogen Detection in Acellular Cerebrospinal Fluid: Diagnostic Insights from a Pediatric Cohort in Colombia)

Dear Prof. Alfonso Bettin Martinez:

Thank you for the privilege of reviewing your work. After receiving feedback from two reviewers, modifications are required before potential publication. Below you will find instructions from the Spectrum editorial office and the reviewer comments.

Revision Guidelines

Sincerely,
Eleanor Powell
Editor
Microbiology Spectrum

Reviewer #1 (Comments for the Author):

Extremely well written-well done. There is a typo on Figure 2 on the table-"Frecuency" rather than "frequency". This study is probably too small to look at outcomes other than length of stay, ICU stay. It seems that there were few cases of bacterial disease, which raises the question of what the neurologic symptoms were that triggered one to do an LP. As noted, the significance of finding HHV 6 requires a nuanced interpretation. There have been questions raised re the same with group B

strep (one of your citations reported a false positive so clinicians must put in correct clinical perspective). Thank you for shining a light on thinking of the absence of pleocytosis as a possible means of immune system evasion.

Reviewer #2 (Comments for the Author):

This study evaluates the traditional diagnostic tools, such as, inflammatory response in the CSF and bacterial and fungal culture, to a molecular syndromic panel in a prospective pediatric cohort of patients suspected of meningitis and/or encephalitis that undergoes lumbar puncture. The study findings are very interesting and raise an important concern on the value of the inflammatory response on screening for molecular syndromic panels. The manuscript is well-written.

Minor comments:

Lines 149-150: This sentence contains results of the present study and should not be in the introduction section. Instead, the authors may refer to studies that had similar findings.

Although the authors have stated that no further confirmatory tests were performed to confirm the results of the molecular syndromic panels, the clinical adjudication should be investigated for each of the 21 clinical cases. In addition, further discussion is needed during the discussion and not only on the limitation paragraph.

Table 3 requires revision of the values on the PPV column, as they are presented as dates instead of numbers.

Title
**Pathogen Detection in Acellular Cerebrospinal Fluid: Diagnostic Insights from a Pediatric Cohort in**
**Colombia**

Short title
Pathogen in Acellular CSF from a Pediatric in Colombia

Authors:

Patiño Patiño, Jhonny Jesús
ORCID: <https://orcid.org/0000-0002-5764-6326>
Correo: jpatinop1@unicartagena.edu.co
Grupo Caribe de Investigación en Enfermedades de Tipo Infeccioso y Resistencia Microbiana. Universidad
Metropolitana. Barranquilla, Colombia. a
Departamento de Medicina Interna. Universidad de Cartagena. Cartagena, Colombia. b

Osorio-Rodríguez, Elber
ORCID: <https://orcid.org/0000-0001-9404-5370>
Correo: osorioelver@gmail.com
Grupo Caribe de Investigación en Enfermedades de Tipo Infeccioso y Resistencia Microbiana. Universidad
Metropolitana. Barranquilla, Colombia. a
El Grupo de Cuidados Intensivos y Atención Integral (GRIMIC). Barranquilla, Colombia. b

García Toscano, Yina Paola
ORCID: <https://orcid.org/0009-0008-7739-3451>
Email: ggarcia@unimetro.edu.co
Grupo Caribe de Investigación en Enfermedades de Tipo Infeccioso y Resistencia Microbiana. Universidad
Metropolitana. Barranquilla, Colombia.

Filott Tamara, Margarita
ORCID:
Email: mafilott@unimetro.edu.co
Grupo Caribe de Investigación en Enfermedades de Tipo Infeccioso y Resistencia Microbiana, Departamento de
Medicina. Universidad Metropolitana. Barranquilla, Colombia.

De-Ávila-Arias Marcio
ORCID: <https://orcid.org/0000-0001-7637-5106>
Email: mdeavilaarias@gmail.com
Grupo de investigación en Biomedicina, Facultad de Ciencias básicas biomédicas. Universidad Metropolitana.
Barranquilla, Colombia.

Rodríguez Sanjuán, Alexander
ORCID: <https://orcid.org/0000-0001-6424-7254>
Correo: directorprogramamedicina@unimetro.edu.co
Grupo de Investigación en Medicina Traslacional. Universidad Metropolitana. Barranquilla, Colombia.

Villarreal-Camacho, Jose Luis:
ORCID:
Email: jvillarrealc@unimetro.edu.co

Grupo de Investigación en Medicina Traslacional, Departamento de Medicina. Universidad Metropolitana.
Barranquilla, Colombia.

Martínez De la Rosa, Walter

ORCID: <https://orcid.org/0000-0003-4106-9357>

Correo: wmartinezdr@unimetro.edu.co

Grupo de investigación en Biomédicina, Facultad de Ciencias básicas biomédicas. Universidad Metropolitana.
Barranquilla, Colombia.

Jorge Bilbao Ramírez

ORCID:

Email: jbilbao55@hotmail.com

Grupo SANUS VIVENTIUM, Departamento de Investigaciones. Universidad Metropolitana. Barranquilla, Colombia.

Coronell-Rodríguez, Wilfrido

ORCID:

Email: wilfridocoronell@unicartagena.edu.co

Facultad de Medicina, Departamento de Pediatría e Infectología. Universidad De Cartagena

# Alfonso Bettin Martínez

ORCID: <https://orcid.org/0000-0002-8335-9929>

Email: abettin@unimetro.edu.co

Corresponding author

Grupo Caribe de Investigación en Enfermedades de Tipo Infeccioso y Resistencia Microbiana. ^a

Grupo de Investigación de Odontología Universidad Metropolitana De Barranquilla (GIOUMEB). ^b

Grupo de Investigación en Medicina Traslacional. ^c

Pathogen Detection in Acellular Cerebrospinal Fluid: Diagnostic Insights from a Pediatric

Cohort in Colombia

89 Pathogen in Acellular CSF from a Pediatric in Colombia

91 Abstract

The diagnosis of pediatric meningoencephalitis has traditionally relied on the detection of
inflammatory markers in cerebrospinal fluid (CSF), particularly pleocytosis, as an indirect indicator of
infection. This assumption presumes that microbial invasion of the central nervous system (CNS)
consistently triggers a measurable immune response. In this prospective observational study
conducted in Cartagena, Colombia, we enrolled 166 pediatric patients with suspected
meningoencephalitis. All patients underwent testing with the FilmArray® Meningitis/Encephalitis (FA-
M/E) panel for direct pathogen detection in CSF, and results were compared to conventional CSF
cytochemical analysis and culture. The FA-M/E panel identified a pathogen in 21 of 166 patients
(12.6%), whereas CSF cultures were negative in all cases. Detected pathogens included 16 viruses
(76.2%), most frequently Enterovirus and Human Herpesvirus 6 (HHV-6) and 5 bacteria (23.8%),
including *Escherichia coli* K1 and *Streptococcus agalactiae*. Remarkably, 100% (21/21) of the FA-
M/E-positive samples exhibited complete absence of pleocytosis (0 white blood cells/ μ L). Additionally,
33.3% (7/21) showed entirely normal CSF biochemistry. Among these cases with a normal CSF
profile, the panel identified two clinically significant bacterial infections and five viral infections that
would have otherwise gone undetected. These findings demonstrate that traditional CSF parameters
are poor predictors of infection in patients with a positive molecular result. Routine reliance on
cytochemical analysis alone may therefore delay diagnosis and treatment. Our results support the
use of molecular diagnostics as a frontline tool, even in the absence of classical inflammatory CSF
markers.

**Importance**

This study challenges the long-standing paradigm that abnormal CSF findings are required to justify
molecular testing in pediatric meningoencephalitis. We show that critical CNS infections can present
with entirely normal CSF profiles, including cases caused by high-risk pathogens such as *E. coli* K1
and *Streptococcus agalactiae*. By highlighting the limitations of pleocytosis and biochemical markers
in detecting early CNS infection, our data underscore the importance of incorporating molecular
diagnostics into routine clinical evaluation, particularly for high-risk pediatric populations. Failure to do
so may result in missed or delayed diagnoses and suboptimal treatment.

**Keywords:** Meningitis, FilmArray, Cerebrospinal Fluid, Pleocytosis, *E. coli* K1, *S. agalactiae*.

**Introduction**

Meningoencephalitis (ME), an inflammatory disease of the central nervous system (CNS), carries a
heavy burden of mortality and long-term neurological sequelae, particularly in the pediatric population

(World Health Organization, 2025) Survivors often face lifelong disabilities, including hearing loss,
cognitive impairment, and motor deficits, imposing a devastating toll on patients and healthcare
systems(Schiess *et al.*, 2021). In Latin America, despite vaccination efforts, bacterial meningitis
remains a significant threat, compounded by phenomena such as serotype replacement in
*Streptococcus pneumoniae* and rising antimicrobial resistance (Villalpando-Carrión *et al.*, 2024). The
cornerstone of ME diagnosis has historically been the analysis of cerebrospinal fluid (CSF), where the
presence of inflammatory markers such as pleocytosis, elevated protein, and low glucose is
interpreted as a proxy for microbial infection (Tunkel *et al.*, 2004). This diagnostic paradigm, however,
assumes that microbial invasion of the CNS invariably triggers a robust and immediate inflammatory
response. This assumption overlooks the complex molecular interplay between pathogen virulence
factors and the host's innate immune response. Conventional diagnostic methods, such as bacterial
culture, are not only slow but also suffer from low sensitivity, a problem exacerbated by the common
and necessary practice of administering empirical antibiotics prior to lumbar puncture (Nigrovic *et al.*,
2008).

In our study, this limitation was starkly evident, with a 0% positivity rate for CSF cultures, even in
cases where bacterial pathogens were confirmed by molecular methods. The advent of syndromic
multiplex PCR panels, such as the FilmArray® Meningitis/Encephalitis (FA-M/E) panel, represents a
technological leap, enabling the direct and rapid detection of pathogen nucleic acids from a small
CSF volume (Leber *et al.*, 2016). This technology allows us to bypass the reliance on the host's
secondary inflammatory response and directly interrogate the presence of the microbe itself. While
the clinical utility of these panels is increasingly recognized, their findings also present an opportunity
to investigate fundamental questions of molecular pathogenesis *in vivo*.

Recent pediatric studies suggests a clinical mismatch between the presence of a pathogen in the
CNS and the classic CSF findings of meningitis (Mizuno *et al.*, 2024).This phenomenon, often termed
acellular or paucicellular meningitis, may occur when pathogens employ sophisticated molecular
strategies to invade the CNS while evading or delaying detection by the host immune system (Kim,

2008a). This is particularly relevant in neonates and infants, whose immature immune systems may mount atypical responses (Camacho-Gonzalez *et al.*, 2013). If clinical decisions, including the use of advanced molecular diagnostics, are gated by traditional inflammatory markers, a significant number of infections may be missed during a critical therapeutic window.

This study employs the high sensitivity of the FA-M/E panel in a prospective pediatric cohort in Cartagena, Colombia, to examine this clinical mismatch between pathogen presence and inflammatory response in the CSF. Our goal was to determine how this molecular tool performs in clinical practice and whether it can reveal infections that escape traditional diagnostic markers, underscoring its value as a first-line approach in the evaluation of suspected CNS infections.

Results

Patient Demographics and FilmArray® Panel Yield

A final cohort of 166 pediatric patients with suspected CNS infection was enrolled (figure 1). The median age was 1.0 year (IQR: 27.3 days to 1.64 years), with 51.8% of patients being infants (≤ 1 year old). The FA-M/E panel identified a pathogen in 21 of 166 samples (12.6%). In stark contrast, standard bacterial and fungal CSF cultures were negative in 100% of the 166 samples, including all 21 cases that were positive by the FA-M/E panel. Of the 21 positive detections, 16 (76.1%) were viral and 5 (23.8%) were bacterial. No fungal pathogens were detected. The most frequently identified pathogens were Enterovirus and Human Herpesvirus 6 (HHV-6). Two cases of co-detection were observed: *Streptococcus agalactiae* with HHV-6, and *Escherichia coli* K1 with Enterovirus. The distribution of detected microorganisms is detailed in Figure 2.

Figure 1. Flowchart of patient inclusion and classification.

Figure 2. Frequency of Microorganisms Detected by the FilmArray M/E Panel (n=21 Positive

187 Samples)

[revised manuscript text omitted]

(\geq 75 mg/dL)
- • Protein: normal (20–80 mg/dL), hypoproteinorrhachia (\leq 20 mg/dL), hyperproteinorrhachia

(≥ 80 mg/dL)

Patients were stratified by FA-M/E result (positive vs. negative). Positive cases were further
subclassified by pathogen type (bacterial, viral, or mixed). We evaluated individual CSF markers and
selected combinations for their association with, and predictive value for, a positive FA-M/E result, as
well as their impact on clinical decision-making.

452 **Statistical Analysis**

Continuous variables were assessed for normality via the Shapiro–Wilk test; non-parametric data are
presented as medians and interquartile ranges (IQR, 25th–75th percentile) and compared using the
Mann–Whitney U test. Categorical variables are expressed as counts and percentages, with group
comparisons conducted using the Chi-square or Fisher’s exact test, as appropriate. Diagnostic
performance metrics sensitivity, specificity, positive predictive value (PPV), and negative predictive
value (NPV) were calculated for each CSF marker and combination, using the FA-M/E panel result as
the reference standard. A two-tailed $p < 0.05$ was considered statistically significant. All analyses
were performed in STATA v17.0 (StataCorp, College Station, TX, USA). Reporting adheres to the
STROBE (Strengthening the Reporting of Observational Studies in Epidemiology) guidelines.

463 **Funding.**

This study was designed by Universidad Metropolitana, Colombia and was funded by BioFire Diagnostics
RCA# BFD-ME-18-004.

467 **Conflict of interest**

468 W.C.R is a Consultant for BioMérieux Col. and has received research funding for other studies not
related to this work. No other authors have reported conflicts of interest.

Acknowledgements

We thank the dedicated professionals at the study site, without whom this work would have not been
possible. Special thanks goes to Ines Martínez, Herando Pinzon, Keterine Palacio del Hospital Infantil
Franco Napoleon Pareja. Casa del Niño.

References

- 1. Agut, H., Bonnafous, P., and Gautheret-Dejean, A. (2015) Laboratory and Clinical Aspects of Human
Herpesvirus 6 Infections. *Clin Microbiol Rev* **28**: 313
<https://pmc.ncbi.nlm.nih.gov/articles/PMC4402955/>. Accessed June 30, 2025.
- 2. Agut, H., Bonnafous, P., and Gautheret-Dejean, A. (2016) Human Herpesviruses 6A, 6B, and
7. *Microbiol Spectr* **4** <https://journals.asm.org/doi/10.1128/microbiolspec.dmiH2-0007-2015>.
Accessed June 30, 2025.
- 3. Boudet, A., Pantel, A., Carles, M.J., Boclé, H., Charachon, S., Enault, C., *et al.* (2019) A review
of a 13-month period of FilmArray Meningitis/Encephalitis panel implementation as a first-line
diagnosis tool at a university hospital. *PLoS One* **14**: e0223887
<https://journals.plos.org/plosone/article?id=10.1371/journal.pone.0223887>. Accessed July 1,
2025.
- 4. Camacho-Gonzalez, A., Spearman, P.W., and Stoll, B.J. (2013) Neonatal Infectious Diseases:
Evaluation of Neonatal Sepsis. *Pediatr Clin North Am* **60**: 367–389.
- 5. Cárdenas Hernández, G.A., Vega, O.R., Castro, M.I., Fleury, A., Gómez Amador, J.L., and
Soto Hernández, J.L. (2009) Cryptococcal Choroid Plexitis an Uncommon Fungal Disease.
Case Report and Review. *Canadian Journal of Neurological Sciences* **36**: 117–122
[https://www.cambridge.org/core/journals/canadian-journal-of-neurological-
sciences/article/cryptococcal-choroid-plexitis-an-uncommon-fungal-disease-case-report-and-
review/676D6FC89DF6A9BCC7C0220F0E435FC9](https://www.cambridge.org/core/journals/canadian-journal-of-neurological-sciences/article/cryptococcal-choroid-plexitis-an-uncommon-fungal-disease-case-report-and-review/676D6FC89DF6A9BCC7C0220F0E435FC9). Accessed July 1, 2025.
- 6. Chen, C.-S., Yao, Y.-C., Lin, S.-C., Lee, Y.-P., Wang, Y.-F., Wang, J.-R., *et al.* (2007)

- Retrograde Axonal Transport: a Major Transmission Route of Enterovirus 71 in Mice. *J Virol*
**81**: 8996–9003 <https://journals.asm.org/doi/10.1128/jvi.00236-07>. Accessed June 30, 2025.
- 7. Djukic, M., Lange, P., Erbguth, F., and Nau, R. (2022) Spatial and temporal variation of routine
parameters: pitfalls in the cerebrospinal fluid analysis in central nervous system infections. *J*
*Neuroinflammation* **19**: 174 <https://pmc.ncbi.nlm.nih.gov/articles/PMC9258096/>. Accessed July
1, 2025.
- 8. Doran, K.S., and Nizet, V. (2004) Molecular pathogenesis of neonatal group B streptococcal
infection: no longer in its infancy. *Mol Microbiol* **54**: 23–31
<https://onlinelibrary.wiley.com/doi/full/10.1111/j.1365-2958.2004.04266.x>. Accessed June 30,
2025.
- 9. Herold, R., Schroten, H., and Schwerk, C. (2019) Virulence Factors of Meningitis-Causing
Bacteria: Enabling Brain Entry across the Blood–Brain Barrier. *International Journal of*
*Molecular Sciences* 2019, Vol 20, Page 5393 **20**: 5393 [https://www.mdpi.com/1422-](https://www.mdpi.com/1422-0067/20/21/5393/htm)
[0067/20/21/5393/htm](https://www.mdpi.com/1422-0067/20/21/5393/htm). Accessed July 1, 2025.
- 10. Houlihan, C.F., Bharucha, T., and Breuer, J. (2019) Advances in molecular diagnostic testing
for central nervous system infections. *Curr Opin Infect Dis* **32**: 244–250
<https://pubmed.ncbi.nlm.nih.gov/30950854/>. Accessed June 30, 2025.
- 11. Kaur, C., Rathnasamy, G., and Ling, E.A. (2016) The Choroid Plexus in Healthy and Diseased
Brain. *J Neuropathol Exp Neurol* **75**: 198–213 <https://dx.doi.org/10.1093/jnen/nlv030>.
Accessed July 1, 2025.
- 12. Kharbat, A.F., Lakshmi-Narasimhan, M., Bhaskaran, S., and Parat, S. (2022) Incidental
Detection of Human Herpesvirus-6 in Cerebrospinal Fluid Analysis: To Treat or Not to Treat?
*Cureus* **14** <https://pubmed.ncbi.nlm.nih.gov/35785001/>. Accessed June 30, 2025.
- 13. Kim, K.S. (2008a) Mechanisms of microbial traversal of the blood–brain barrier. *Nature*
*Reviews Microbiology* 2008 6:8 **6**: 625–634 <https://www.nature.com/articles/nrmicro1952>.
Accessed June 30, 2025.

- 14. Kim, K.S. (2008b) Mechanisms of microbial traversal of the blood–brain barrier. *Nature*
*Reviews Microbiology 2008 6:8 6*: 625–634 <https://www.nature.com/articles/nrmicro1952>.
Accessed June 30, 2025.
- 15. Kim, K.S. (2016) Human Meningitis-Associated Escherichia coli . *EcoSal Plus 7*
<https://journals.asm.org/doi/10.1128/ecosalplus.esp-0015-2015>. Accessed June 30, 2025.
- 16. King, M.R., Steenbergen, S.M., and Vimr, E.R. (2007) Going for baroque at the Escherichia
coli K1 cell surface. *Trends Microbiol 15*: 196–202
<https://www.cell.com/action/showFullText?pii=S0966842X07000492>. Accessed June 30, 2025.
- 17. Leber, A.L., Everhart, K., Balada-Llasat, J.M., Cullison, J., Daly, J., Holt, S., *et al.* (2016)
Multicenter evaluation of biofire filmarray meningitis/encephalitis panel for detection of
bacteria, viruses, and yeast in cerebrospinal fluid specimens. *J Clin Microbiol 54*: 2251–2261
<https://journals.asm.org/doi/10.1128/jcm.00730-16>. Accessed June 30, 2025.
- 18. Liu, Q., and Long, J.E. (2025) Insight into the Life Cycle of Enterovirus-A71. *Viruses 2025, Vol*
*17, Page 181 17*: 181 <https://www.mdpi.com/1999-4915/17/2/181/htm>. Accessed June 30,
2025.
- 19. Majer, A., McGreevy, A., and Booth, T.F. (2020) Molecular Pathogenicity of Enteroviruses
Causing Neurological Disease. *Front Microbiol 11*: 514822.
- 20. Megli, C.J., Carlin, S.M., Giacobe, E.J., Hillebrand, G.H., and Hooven, T.A. (2025) Virulence
and pathogenicity of group B Streptococcus: Virulence factors and their roles in perinatal
infection. *Virulence 16* <https://pubmed.ncbi.nlm.nih.gov/39844743/>. Accessed June 30, 2025.
- 21. Mizuno, S., Kusama, Y., Otake, S., Ito, Y., Nozaki, M., and Kasai, M. (2024) Epidemiology of
pediatric meningitis and encephalitis in Japan: a cross-sectional study. *Microbiol Spectr 12*
<https://journals.asm.org/doi/10.1128/spectrum.01192-24>. Accessed June 30, 2025.
- 22. Nigrovic, L.E., Kuppermann, N., Macias, C.G., Cannavino, C.R., Moro-Sutherland, D.M.,
Schremmer, R.D., *et al.* (2007) Clinical prediction rule for identifying children with
cerebrospinal fluid pleocytosis at very low risk of bacterial meningitis. *JAMA 297*: 52–60

- <https://pubmed.ncbi.nlm.nih.gov/17200475/>. Accessed June 30, 2025.
- 23. Nigrovic, L.E., Malley, R., Macias, C.G., Kanegaye, J.T., Moro-Sutherland, D.M., Schremmer,
R.D., *et al.* (2008) Effect of Antibiotic Pretreatment on Cerebrospinal Fluid Profiles of Children
With Bacterial Meningitis. *Pediatrics* **122**: 726–730 /pediatrics/article/122/4/726/71330/Effect-
of-Antibiotic-Pretreatment-on-Cerebrospinal. Accessed June 30, 2025.
- 24. Precit, M.R., Yee, R., Pandey, U., Fahit, M., Pool, C., Naccache, S.N., and Barda, J.D. (2020)
Cerebrospinal Fluid Findings Are Poor Predictors of Appropriate FilmArray
Meningitis/Encephalitis Panel Utilization in Pediatric Patients. *J Clin Microbiol* **58**
<https://pubmed.ncbi.nlm.nih.gov/31852767/>. Accessed June 30, 2025.
- 25. Prusty, B.K., Krohne, G., and Rudel, T. (2013) Reactivation of Chromosomally Integrated
Human Herpesvirus-6 by Telomeric Circle Formation. *PLoS Genet* **9**: e1004033
<https://journals.plos.org/plosgenetics/article?id=10.1371/journal.pgen.1004033>. Accessed June
30, 2025.
- 26. Ramchandrar, N., Coufal, N.G., Warden, A.S., Briggs, B., Schwarz, T., Stinnett, R., *et al.* (2021)
Metagenomic Next-Generation Sequencing for Pathogen Detection and Transcriptomic
Analysis in Pediatric Central Nervous System Infections. *Open Forum Infect Dis* **8**
<https://pubmed.ncbi.nlm.nih.gov/34104666/>. Accessed June 30, 2025.
- 27. Rose, R., Häuser, S., Stump-Guthier, C., Weiss, C., Rohde, M., Kim, K.S., *et al.* (2018)
Virulence factor-dependent basolateral invasion of choroid plexus epithelial cells by pathogenic
*Escherichia coli* in vitro. *FEMS Microbiol Lett* **365** <https://dx.doi.org/10.1093/femsle/fny274>.
Accessed July 1, 2025.
- 28. Schiess, N., Groce, N.E., and Dua, T. (2021) The Impact and Burden of Neurological Sequelae
Following Bacterial Meningitis: A Narrative Review. *Microorganisms* **2021**, Vol 9, Page 900 **9**:
900 <https://www.mdpi.com/2076-2607/9/5/900/htm>. Accessed June 30, 2025.
- 29. Schnuriger, A., Vimont, S., Godmer, A., Gozlan, J., Gallah, S., Macé, M., *et al.* (2022)
Differential Performance of the FilmArray Meningitis/Encephalitis Assay To Detect Bacterial

- and Viral Pathogens in Both Pediatric and Adult Populations. *Microbiol Spectr* **10**: e02774-21
<https://pmc.ncbi.nlm.nih.gov/articles/PMC9045182/>. Accessed July 1, 2025.
- 30. Schwerk, C., Tenenbaum, T., Kim, K.S., and Schroten, H. (2015) The choroid plexus—a multi-
role player during infectious diseases of the CNS. *Front Cell Neurosci* **9**: 132238
www.frontiersin.org. Accessed July 1, 2025.
- 31. Tunkel, A.R., Hartman, B.J., Kaplan, S.L., Kaufman, B.A., Roos, K.L., Scheld, W.M., and
Whitley, R.J. (2004) Practice Guidelines for the Management of Bacterial Meningitis. *Clinical*
*Infectious Diseases* **39**: 1267–1284 <https://dx.doi.org/10.1086/425368>. Accessed June 30,
2025.
- 32. Villalpando-Carrión, S., Henao-Martínez, A.F., and Franco-Paredes, C. (2024) Epidemiology
and Clinical Outcomes of Bacterial Meningitis in Children and Adults in Low- and Middle-
Income Countries. *Curr Trop Med Rep* **11**: 60–67
<https://link.springer.com/article/10.1007/s40475-024-00316-0>. Accessed June 30, 2025.
- 33. World Health Organization (2025) Meningitis. *WHO* 1 [https://www.who.int/news-room/fact-](https://www.who.int/news-room/fact-sheets/detail/meningitis)
[sheets/detail/meningitis](https://www.who.int/news-room/fact-sheets/detail/meningitis). Accessed June 30, 2025.
- 34. Yamayoshi, S., Fujii, K., and Koike, S. (2014) Receptors for enterovirus 71. *Emerg Microbes*
*Infect* **3** <https://www.tandfonline.com/doi/abs/10.1038/emi.2014.49>. Accessed June 30, 2025.
- 35. Guerrero, M.P., Romero, A.F., Luengas, M., Dávalos, D.M., Mesa-Monsalve,
592 J.G., Vivas-Trochez, R., *et al.*(2022) Etiology and Risk Factors for Admission to
593 the Pediatric Intensive Care Unit in Children With Encephalitis in a Developing
Country. *Pediatric Infectious Disease Journal* **41**: 806–812
[https://journals.lww.com/pidj/fulltext/2022/10000/etiology_and_risk_factors_for_a](https://journals.lww.com/pidj/fulltext/2022/10000/etiology_and_risk_factors_for_a_dmission_to_the.4.aspx)
[dmission_to_the.4.aspx](https://journals.lww.com/pidj/fulltext/2022/10000/etiology_and_risk_factors_for_a_dmission_to_the.4.aspx). Accessed July 2, 2025.

Figure 1. Flowchart of patient inclusion and classification. Patients with clinical suspicion of meningitis or encephalitis were initially screened. After lumbar puncture, cerebrospinal fluid (CSF) samples were evaluated for eligibility. Exclusion criteria included recent neurosurgery, traumatic brain injury, repeated samples, or incomplete clinical data. A total of 166 patients were included and underwent analysis using the FilmArray® Meningitis/Encephalitis (FA-ME) panel. Based on the FA-ME results, patients were categorized as positive (n=21) or negative (n=145).

Variables	Total (n = 166)	FilmArray Positive (n = 21)	FilmArray Negative (n = 145)	p-value
Sociodemographic, n (%)				
Male	94 (56.6)	10 (47.6)	84 (57.9)	0.373 ^a
Female	72 (43.4)	11 (52.4)	61 (42.1)	
≤ 1 year	86 (51.8)	12 (57.1)	74 (51.0)	0.601 ^a
> 1 year	80 (48.2)	9 (42.9)	71 (49.0)	
Signs and Symptoms, n (%)				
Fever	147 (88.6)	20 (95.2)	127 (87.6)	0.472 ^a
Tachycardia	95 (57.2)	8 (38.1)	87 (60.0)	0.069 ^a
Tachypnea	25 (15.1)	5 (23.8)	20 (13.8)	0.322 ^a
Neurological alterations ^{‡, n}	42 (25.3)	6 (28.6)	36 (24.8)	0.712 ^a
Gastrointestinal symptoms	23 (13.9)	3 (14.3)	20 (13.8)	1.000 ^b

Laboratory Admission, (IQR)	at median				
Leukocytes ($\times 10^3/\text{mm}^3$)	10,400 13,600	(7,750– 11,400)	9,000 14,000	(6,200– 10,600)	(7,800– 0.089 ^c)
Neutrophils (%)	54.4 (37–64)	55.3 (37–62.8)	54.0 (37–64)		0.951 ^c
Lymphocytes (%)	36.7 (27.1–48.8)	34.5 (29.8–45.6)	37.2 (27–49)		0.541 ^c
Platelets (No./ mm^3)	352,000 449,750	(258,750– 458,000)	351,000 446,000	(277,000– 250,000–)	0.820 ^c
Complicated, n (%)					
Cardiovascular	9 (5.4)	0 (0)	9 (6.2)		0.605 ^b
Respiratory	26 (15.7)	4 (19.0)	22 (15.2)		0.747 ^b
Renal	3 (1.8)	0 (0)	3 (2.1)		1.000 ^b
Neurological ⁿ	4 (2.4)	1 (4.8)	3 (2.1)		0.421 ^b
Length of Stay, days (IQR)					
Hospitalization	4 (2–7.5)	6 (4–12)	3 (2–7)		0.006 ^c
ICU	7 (4–12)	13 (10.3–18)*	7 (3–11)		0.029 ^c
Clinical Outcomes, n (%)					
Neurological sequelae, n (%) ⁿ	18 (10.8)	2 (9.5)	16 (11.0)		1.000 ^b
Death, n (%)	4 (2.4)	0 (0)	4 (2.8)		1.000 ^b

Table 1. Baseline Demographic, Clinical, Laboratory, and Outcome Characteristics of the Cohort by FilmArray M/E Panel Result. Data are presented as n (%) for categorical variables and median (interquartile range [IQR]) for continuous variables. Group comparisons between FilmArray-positive and -negative patients were conducted using the chi-square test (^a), Fisher's exact test (^b) for categorical variables, and the Mann-Whitney U test (^c) for continuous variables; a p-value < 0.05 was considered statistically significant. ¥ indicates convulsion or Glasgow Coma Scale score ≤ 13; * indicates hydrocephalus. Abbreviations: FA-M/E, FilmArray Meningitis/Encephalitis; ICU, intensive care unit; GCS, Glasgow Coma Scale; IQR, interquartile range; WBC, white blood cells.

Figure 2. Etiological Distribution of Pathogens Detected by the FilmArray M/E Panel. The bar chart illustrates the frequency of each unique pathogen identified across the 21 positive patient samples. Enterovirus and Human Herpesvirus 6 (HHV-6) were the most frequently detected agents. The total number of pathogens identified (n=24) is greater than the number of positive patients (n=21) due to three cases of co-detection (*Streptococcus agalactiae* with HHV-6; *Escherichia coli* K1 with Enterovirus; and Enterovirus with HHV-6).

Variables	Total (n=166)	FilmArray Positive			FilmArray (n=145)	Negative	p-value
		Bacteria (n=4)	Virus (n=15)	Bacteria–Virus (n=2)			
Individual biomarkers, n (%)							
Appearance (abnormal)	11 (6.6)	0	0	0	11 (7.6)		0.362a
Elevated opening pressure	15 (9.0)	0	0	0	15 (10.3)		0.221a
Leukocytes	7 (4.2)	0	0	0	7 (4.8)		0.597a
Normal CSF glucose	120 (72.3)	3 (14.3)	10 (47.6)	1 (4.8)	106 (73.1)		0.538b
Hypoglycorrhachia	28 (16.9)	1 (4.8)	4 (19.0)	1 (4.8)	22 (15.2)		0.130a
Normal CSF protein	82 (49.4)	2 (9.5)	6 (28.6)	0 (0)	74 (51.0)		0.268b
Hyperproteinorrhachia	62 (37.3)	1 (4.8)	8 (38.1)	2 (9.5)	51 (35.2)		0.128b
Hypoproteinorrhachia	22 (13.3)	1 (4.8)	1 (4.8)	0 (0)	20 (13.8)		0.590b
Combined biomarkers, n (%)							
Normal glucose + normal protein	67 (40.4)	2 (9.5)	5 (23.8)	0 (0)	60 (41.4)		0.482b
Hypoglycorrhachia + normal protein	8 (4.8)	0 (0)	1 (4.8)	0 (0)	7 (4.8)		1.000a
Hyperproteinorrhachia + normal glucose	37 (22.3)	0 (0)	5 (23.8)	1 (4.8)	31 (21.4)		0.574a
Normal glucose + hypoproteinorrhachia	16 (9.6)	1 (4.8)	0 (0)	0 (0)	15 (10.3)		0.696a
Hypoglycorrhachia + hypoproteinorrhachia	8 (4.8)	0 (0)	1 (4.8)	0 (0)	7 (4.8)		1.000a
Hypoglycorrhachia + hyperproteinorrhachia	20 (12.0)	1 (4.8)	3 (14.3)	1 (4.8)	15 (10.3)		0.141a

Table 2. Individual and Combined Cerebrospinal Fluid Cytochemical Findings by FilmArray M/E Panel Result. Data are presented as n (%) for each CSF parameter. Comparisons between FilmArray-positive subgroups (bacterial, viral, bacterial-viral) and FilmArray-negative patients were performed using the chi-square test (a) or Fisher's exact test (b); a p-value < 0.05 was considered statistically significant. **Abbreviations & Notes:** CSF, cerebrospinal fluid; FA-M/E, FilmArray Meningitis/Encephalitis; a Chi-square test; b Fisher's exact test.

Impact of FilmArray Result on Length of Stay

Figure 3. Hospital and ICU Length of Stay Stratified by FilmArray M/E Result. Boxplots depict the median and interquartile range (IQR) of total hospital stay (left) and intensive care unit (ICU) stay (right) in days for patients with a positive FilmArray M/E panel (n = 21) versus those with a negative result (n = 145). Whiskers extend to the most extreme data points not considered outliers. Statistical significance between groups was assessed using the Mann–Whitney U test; p-values are indicated above each comparison.

Variables	Sensitivity (TP/TP+FN)%	Specificity (TN/FP+TN)%	Positive Value (TP/TP+FP)%	Predictive Negative Value (TN/TN+FN)%	Predictive			
Individual biomarkers								
Appearance (abnormal)	0/21	0	134/145	92.4	0/11	0	134/155	86.5
Elevated opening pressure	0/21	0	130/145	89.7	0/15	0	130/151	86.1
Leukocytes	0/21	0	138/145	95.2	0/7	0	138/159	86.8
Normal CSF glucose	14/21	66.7	39/145	26.9	14/120	11.7	39/46	84.8
Hypoglycorrhachia	6/21	28.6	123/145	84.8	Jun-28	21.4	123/138	89.1
Normal CSF protein	8/21	38.1	71/145	49	Aug-82	9.8	71/84	84.5
Hyperproteinorrhachia	11/21	52.4	94/145	64.8	Nov-62	17.7	94/104	90.4
Hypoproteinorrhachia	2/21	9.5	125/145	86.2	Feb-22	9.1	125/144	86.8
Combined biomarkers								
Normal glucose normal protein	+ 7/21	33.3	85/145	58.6	Jul-67	10.4	85/99	85.9

Hypoglycorrhachia + normal protein	1/21	4.8	138/145	95.2	1-Aug	12.5	138/158	87.3
Hyperproteinoorrhachia + normal glucose	6/21	28.6	114/145	78.6	Jun-37	16.2	114/129	88.4
Normal glucose hypoproteinoorrhachia	1/21	4.8	130/145	89.7	Jan-16	6.3	130/150	86.7
Hypoglycorrhachia + hypoproteinoorrhachia	1/21	4.8	138/145	95.2	1-Aug	12.5	138/158	87.3
Hypoglycorrhachia + hyperproteinoorrhachia	5/21	23.8	130/145	89.7	May-20	25	130/146	89

Table 3. Diagnostic Performance of Individual and Combined CSF Cytochemical Findings for the FA-M/E Panel. Sensitivity (TP/[TP + FN]), specificity (TN/[FP + TN]), positive predictive value (PPV = TP/[TP + FP]) and negative predictive value (NPV = TN/[TN + FN]) are shown as both fraction and percentage. A p-value < 0.05 was considered statistically significant. **Abbreviations & Notes:** CSF, cerebrospinal fluid; FA-M/E, FilmArray Meningitis/Encephalitis; TP, true positives; TN, true negatives; FP, false positives; FN, false negatives.

Response to reviewers:

Reviewer # 1:

a) There is a typo on Figure 2 on the table-"Frecuency" rather than "frequency".

Response: We already modify the title Y axis on figure 2 with “Frequency”

b) This study is probably too small to look at outcomes other than length of stay, ICU stay.

Response: We agree with the reviewer. We have added text in both the Results and Limitations sections to acknowledge that the small number of FilmArray-positive cases limits the statistical power to evaluate clinical outcomes. These results are now described as exploratory (Results section-clinical outcomes, lines 214–215; Limitations, lines 424–425).

c) It seems that there were few cases of bacterial disease, which raises the question of what the neurologic symptoms were that triggered one to do an LP. As noted.

Response: We have expanded the *Methods* → *Patient Selection* section to clearly describe the neurological and clinical indications that prompted lumbar puncture in our cohort. Specifically, LP was performed in patients presenting with fever plus one or more neurological findings (new-onset seizures, altered level of consciousness, focal neurological signs, or meningeal signs), or in neonates with fever without source according to institutional protocol (see *Methods*, lines 466–467) and Supplementary table S1.

d) the significance of finding HHV 6 requires a nuanced interpretation.

Response: Regarding HHV-6, we have added a paragraph to the *Discussion* emphasizing that the detection of HHV-6 in CSF requires careful clinical interpretation, as the assay cannot distinguish between active infection, reactivation, or chromosomally integrated virus (lines 379–387).

e) There have been questions raised re the same with group B strep (one of your citations reported a false positive so clinicians must put in correct clinical perspective).

Response: For *Group B Streptococcus*, we now acknowledge that previous studies (including one we cited) have reported occasional false-positive FilmArray results. We have expanded the discussion to note that clinicians should interpret GBS detections in the appropriate clinical context and, when feasible, confirm by additional testing (lines 363–368).

Reviewer #2:

a) Lines 149–150: This sentence contains results of the present study and should not be in the introduction section. Instead, the authors may refer to studies that had similar findings.

Response: We thank the reviewer for this observation. We have removed the sentence reporting our own results from the Introduction and replaced it with a general statement supported by references (Leber et al., 2016; Boudet et al., 2019). The sentence has been moved to the results section where the corresponding findings are presented (Introduction, lines 156–157; Results, lines 185–188).

b) Although the authors have stated that no further confirmatory tests were performed to confirm the results of the molecular syndromic panels, the clinical adjudication should be investigated for each of the

21 clinical cases. In addition, further discussion is needed during the discussion and not only on the limitation paragraph.

Response: We agree. We have conducted a retrospective clinical adjudication for all 21 FilmArray-positive cases, classifying them as definite, probable, possible, or not infection. The methods and criteria are now detailed in the *Methods* section, and a summary of the adjudication is provided in Supplementary Table S1. We have also expanded the *Discussion* to emphasize the clinical interpretation of molecular detections (*Discussion*, lines 314–317).

c) Table 3 requires revision of the values on the PPV column, as they are presented as dates instead of numbers

Response: We thank the reviewer for noticing this formatting error. The PPV column has been corrected to display numerical values (percentages) rather than dates, and all diagnostic performance parameters have been recalculated and verified (Table 3).

Re: Spectrum02122-25R1 (Pathogen Detection in Acellular Cerebrospinal Fluid: Diagnostic Insights from a Pediatric Cohort in Colombia)

Dear Prof. Alfonso Bettin Martinez:

I'm pleased to inform you that your manuscript has been accepted, and I am forwarding it to the ASM production staff for publication. Your paper will first be checked to make sure all elements meet the technical requirements. ASM staff will contact you if anything needs to be revised before copyediting and production can begin. Otherwise, you will be notified when your proofs are ready to be viewed.

Sincerely,
Eleanor Powell
Editor
Microbiology Spectrum